# Research on Operation Efficiency and Membrane Fouling of A^2^/O-MBR in Reclaimed Water Treatment

**DOI:** 10.3390/membranes9120172

**Published:** 2019-12-17

**Authors:** Fenfen Li, Xinyue An, Cuimin Feng, Jianwei Kang, Junling Wang, Hongying Yu

**Affiliations:** 1Key Laboratory of Urban Stormwater System & Water Environment of MOE, Beijing University of Civil Engineering and Architecture, Beijing 100044, China; 18130657649@163.com (F.L.); wangjunling@bueca.edu.cn (J.W.); 2National Demonstration Center for Experimental Water Environment Education, Beijing University of Civil Engineering and Architecture, Beijing 100044, China; Theozyam_yhy@163.com; 3The Logistic Service Group, Beijing Language and Culture University, Beijing 100083, China; anxinyue_home@126.com; 4China Railway Construction Group Co., Ltd., Beijing 100040, China; 13051828193@163.com

**Keywords:** A^2^/O-MBR, reclaimed water, operation efficiency, membrane, fouling

## Abstract

Taking the public building domestic wastewater as an example, the combination of the MBR (membrane bioreactor) process and the traditional A^2^/O (anaerobic-anoxic-oxic) process was established and analyzed in terms of the removal effect of the pollutants, the impact of the microbial community changes on the process, the MBR membrane fouling, the cleaning methods, and the cleaning performance. The results indicated that the effluent water quality of the domestic wastewater treated with the A^2^/O-MBR process was stable and met the emission requirement to the natural water body. There was good microbial diversity in raw water, the anaerobic tank, the anoxic tank, the aerobic MBR tank and the disinfection tank, and the aerobic MBR tank has a wide variety of aerobic microorganisms, which elevates the removal of organics and the nitrification of ammonia nitrogen and ensures the qualification of nitrogen and phosphorus indexes of the system effluent water. For the fouled membrane, the surface of the contaminated membrane was covered by macromolecular contaminants, causing the membrane flux to drop, and after different cleaning methods to the membrane were compared, it was discovered that the combined use of cleaning agents had better effects than the single ones, and the cleaning method of sodium hydroxide followed by hydrochloric acid showed the best effect, achieving a membrane flux restoration ratio above 80% after cleaning.

## 1. Introduction

Continuous expansion of the population in cities can lead to a water crisis, such as in Beijing, a typical megalopolis with a serious shortage of water resources; per capita water resources has reduced to about 100 m^3^ [1,2]. To improve wastewater recycling efficiency, the Beijing government has formulated a series of policy documents and put forward mandatory requirements that the wastewater treatment rate must reach 90% or above via the upgrade of the existing and new reclaimed water plants [3].

A^2^/O is a wastewater treatment process developed by American and South African experts in the 1970s based on an anaerobic–aerobic process [4]. It is a simple process with the benefits of simultaneous nitrogen and phosphorus removal and short total hydraulic retention time and has been widely used in the wastewater treatment plants. However, its nitrogen removal efficiency is facing a bottleneck for further improvement. Therefore, Adam et al. [5] introduced the A^2^/O-MBR (membrane bioreactor, MBR) process in 2002, which combines the A^2^/O process and the MBR process.

In recent years, the wastewater treatment and regeneration processes based on membrane technology have been developing rapidly, indicating good potential and competitiveness [6]. The membrane bioreactor (MBR) was proposed by Smith in 1969 [7]. Compared with the traditional treatment, MBR has the benefits of high effluent water quality, small footprint, low excess activated sludge and high automation, which promoted the establishment and large-scale application of the A^2^/O-MBR process in the nitrogen and phosphorus removal of heavy-duty wastewater treatment [8]. The A^2^/O-MBR process not only improves the effluent water quality in wastewater treatment and regeneration but also has the benefits of easy upgrading and low engineering quantity [9,10]. 

In the A^2^/O-MBR process, the addition of a MBR unit can create conditions for simultaneous nitrification and denitrification to improve the nitrogen removal rate. The removal of phosphorus in the process mainly relies on the excessive absorption of dissolved orthophosphate and the discharge of MBR membrane phosphorus-rich sludge in the aerobic tank [11]. However, the membrane fouling caused by extracellular polymers (EPS) and soluble microbial products (SMPs) is very serious, so the membrane needs to be cleaned. At present, the commonly used chemical agents for membrane cleaning include acidic cleaning agents, alkaline cleaning agents, oxidizing agents, complexing agents and detergents [12].

The removal of nitrogen relies on the nitrification and denitrification of aerobic and anoxic tanks and the presence of membrane modules; the microbial population can continue to survive and improve the system nitrification efficiency. The removal of phosphorus relies on the dissolved orthophosphate being absorbed by the phosphate-accumulating bacteria to form a phosphorus-rich sludge, and finally achieving phosphorus removal by discharging the excess sludge out of the system. The long-term use of the membrane module will lead to membrane fouling, which leads to a decrease of membrane flux and a decrease of processing capacity or an increase in costs. Therefore, it is necessary to study the membrane fouling characteristics and cleaning methods of the A^2^/O-MBR process, which is of great significance for the long-term stable use of the MBR [11,12].

In this paper, taking the treatment of public building domestic wastewater as an example, the focus is set on the removal effect of pollutants with the A^2^/O-MBR process, the impact of the microbial community changes on the process, the MBR membrane fouling, the cleaning methods, and the cleaning performance of the membrane. It also provides theoretical support for the future upgrading projects of wastewater treatment plants.

## 2. Materials and Methods 

### 2.1. Raw Water

With reference to the influent water quality of multiple wastewater treatment plants in Beijing [13,14], a certain proportion of CH_3_COONa·3H_2_O (China Lab Purchasing Mall, Beijing, China), C_6_H_12_O_6_ (China Lab Purchasing Mall, Beijing, China), KH_2_PO_4_ (China Lab Purchasing Mall, Beijing, China), and NH_4_Cl (China Lab Purchasing Mall, Beijing, China) were mixed to prepare raw water, and the water quality is given in Table 1.

### 2.2. Experimental Device

The device of the A^2^/O-MBR (Beijing Great Wall Electronic Equipment Co., Ltd, Beijing, China) process is shown in Figure 1. The system consists of the inlet tank, the anaerobic tank, the anoxic tank, the aerobic membrane tank and the disinfection tank. The raw water enters the anaerobic, anoxic, and aerobic membrane tanks in turn, and the mixture from the aerobic tank returns to the anoxic tank through the sludge return pump, with the reflux ratio at 200% to 300%. A stirrer is arranged in the anoxic tank and an aeration device, with multiple functions such as oxygen supply, mixing and scrubbing so as to remove organics, is provided in the aerobic membrane tank. In this system, nitrification and denitrification conditions can be provided for nitrogen removal, and anaerobic excess phosphorus release and aerobic phosphorus uptake conditions can be provided for phosphorus removal. Under the periodic operation of the self-priming pump, the biochemical treatment effluent enters the disinfection tank during the 8 min of negative pressure, while during the other 2-min standby of the pump, there is the separation of the activated sludge from the water for purification in the plate membrane tank, followed by disinfection with sodium hypochlorite. The system is equipped with the time relays, liquid level feedback control meters, flow meters, and other electrical instruments, so as to be controlled automatically.

After the test device was built, the first step was to inoculate and acclimate the sludge. The prepared wastewater and the concentrated sludge taken from a wastewater treatment station of a service industry in Beijing were added to the reactor. After a day of aeration, the sewage is allowed to settle for 2 h, then 20% supernatant is removed and then an equal proportion of wastewater is added. When the MLSS (mixed liquid suspended solids) concentration increases steadily, it indicates that the activated sludge is adapted to the sewage environment. At this time, the MBR membrane module can be put into the reactor to start the operation of the device. After 15 d, when the removal rate of COD_Cr_ (chemical oxygen demand) and NH_4_^+^-N reaches 80%, it indicates that the sludge has been domesticated and can be used for further experimental study.

The operation parameters of the device are shown in Table 2.

### 2.3. MBR Membrane Module

The membrane module in the MBR tank was made of polyvinylidene fluoride (PVDF) (Jiangsu Jiayi Xinchen Membrane Technology Co., Ltd., Yixing, China)with a pore size of 0.20 μm. The immersed plate microfiltration membrane was used in the MBR membrane tank, including the baffle, membrane supporting material, and polymer membrane. The specific performance parameters of the plate membrane are shown in Table 3.

The long-term use of membrane modules can lead to membrane fouling, which in turn leads to the decrease of membrane flux and the decrease of processing capacity or the increase of costs, which is an important factor limiting the development of MBR process. Therefore, in order to test the effect of different cleaning methods, the membrane flux was detected using an ultrafiltration cup. See Equation (1).
(1)$$J=\frac{V}{A\cdot T}$$
where *J* is membrane flux (L/m^2^·h), *V* is sampling volume (L), *A* is the effective area of the ultrafiltration membrane (m^2^), *T* is sampling time (h).

### 2.4. Analysis Methods of Water Quality

Routine water quality parameters: The water quality parameters such as COD_Cr_, BOD_5_ (biochemical oxygen demand), NH_4_^+^-N, TN (total nitrogen), and TP (total phosphorus) were tested according to the methods provided in the *Chinese Water and Wastewater Monitoring Methods, 4th ed* [15], and the final value of each parameter was to take water detection in the upper, middle, and lower parts of each reactor, and then take the average of the three values.

Microbial community structure analysis: The microbial DNA was extracted from the activated sludge samples sampled in the anaerobic tank, the anoxic tank, the aerobic MBR membrane tank, and the disinfection tank. After successful verification, the V3-V4 sections of the 16S rRNA was amplified with PCR (Beijing Weihui Biological Technology Co., Ltd., Beijing, China), followed by mixing sample, library building, and distinguishing the sample with the preset TAG sequence. After the qualification of the DNA library, the samples were sequenced with the Illumina Hiseq 2500 PE250 high-throughput sequencing platform. The microbial diversity of the samples was analyzed based on the sequencing results.

X-ray energy spectrum analysis (EDS) (Hitachi, Tokyo, Japan): The samples on the films were coated with gold for 5–10 min with a sputter coating system, and the coated samples were observed under a cold field emission high-resolution scanning electron microscope (HITACHI SU8220, Hitachi, Tokyo, Japan).

## 3. Results and Discussion

### 3.1. Water Quality Treatment Effects

#### 3.1.1. Organics

Figure 2 shows the changes of COD_Cr_ and BOD_5_ along with the process of the anaerobic tank, the anoxic tank, the aerobic MBR tank and the disinfection tank. The COD_Cr_ of influent was between 274.7 mg/L and 325.3 mg/L, the effluent was stably below 30 mg/L, and the total removal rate was over 90.6%. The total removal rate of BOD_5_ was 88.8%. The effluent water met the Class IV requirements of COD_Cr_ ≤ 30 mg/L, BOD_5_ ≤ 6 mg/L in the *Environmental Quality Standards for Surface Water*. The removal of COD_Cr_ was mainly contributed by the aerobic tank (removed 40%), just due to the biodegradation of the aerobic microorganisms, the efficient interception, and prolonged sludge age provided by the membrane module. BOD_5_ behaved in the same trend along with COD_Cr_.

#### 3.1.2. Nitrogen

Figure 3 shows the changes in nitrogen along the process. The removal rate of NH_4_^+^-N is 87% and mainly occurs in the aerobic unit. The ammonia nitrogen was nitrified by nitrifying bacteria, further decomposed and oxidized to nitrate nitrogen, and after being refluxed to the anoxic tank, the denitrification process completed the formation of nitrate nitrogen, and the concentration of NO_3_-N in the anoxic unit increased. TN was removed at each stage of the reaction but was mainly concentrated in the anaerobic and aerobic tanks. The anaerobic section was mainly effected by microbial metabolism, anaerobic ammonium oxidation, precipitation, and adsorption. The nitrification reaction mainly occurred in the aerobic section, which greatly reduced the concentration of NH_4_^+^-N and stabilized the effluent water concentration at about 1 mg/L, meeting the Class IV requirements of NH_4_^+^-N ≤ 1.5 mg/L in the *Environmental Quality Standards for Surface Water*.

The traditional biological nitrogen removal process is mainly based on microbial nitrification and denitrification. In an aerobic environment, nitrifying bacteria oxidize NH_4_^+^-N to NO_2_-N and NO_3_-N through nitrification. For denitrification in an anoxic environment; the bacteria reduce NO_2_-N and NO_3_-N to N_2_ by denitrification. When the MBR unit was added to the system, the oxygen mass transfer process was blocked due to the high sludge concentration in the reactor, so that the sludge flocs exist in the anoxic environment and conditions were created for the occurrence of simultaneous nitrification and denitrification. Therefore, from the anoxic tank to the aerobic membrane tank, the removal rate of TN was relatively high.

#### 3.1.3. Total Phosphorus

Figure 4 shows the changes of TP along the process. At the anaerobic section, the TP of the wastewater increased significantly. Under the action of anaerobic bacteria, some easily degradable macromolecular organics are transformed into volatile fatty acids (VFA). With sufficient organics, the phosphate-accumulating bacteria absorb the VFA and synthesize the PHB (poly-β-hydroxybutyrate) stored in-vivo, while orthophosphate is released into the mixture. The released energy is used to maintain the bacteria alive in the anaerobic environment, resulting in an increase in the concentration of dissolved phosphorus and a decrease in the concentration of organics in the mixture. The concentration of TP in the subsequent anoxic tank decreases in a lesser manner. In the aerobic tank, the concentration of organics in the mixture is lower, the phosphate-accumulating bacteria rely on the PHB stored in-vivo to obtain energy and excessively absorb the dissolved orthophosphate, so predominantly contributing to the phosphorus uptake, and phosphorus removal can be achieved via the discharge of phosphorus-rich sludge into the MBR membrane tank. A significant decline of TP in the aerobic MBR tank was accounted for but the concentration was not steady through the local anaerobic environment where the phosphate-accumulating bacteria performed the anaerobic phosphorus release, but such anaerobic microenvironment might exist as a result of the blockage of the oxygen transfer by the mud layer [16,17]. Therefore, in engineering applications, the high phosphorus concentration of effluent water can be reduced by the addition of the phosphorus removing agents to meet the requirement of the *Environmental Quality Standards for Surface Water*.

In summary, the effluent water of the A^2^/O-MBR process can basically meet the stringent requirements of water emission in the parameters of organics, nitrogen, and phosphorus.

### 3.2. Microbial Community in the System

#### 3.2.1. Diversity Analysis

Microorganisms are the primary player in wastewater treatment, and the diversity of the community greatly influences the performance of wastewater treatment systems. Five water samples were tested, numbered as S1 to S5, as the representatives of the influent, the anaerobic tank, the anoxic tank, the aerobic MBR tank, and the effluent of the disinfection tank, respectively. The PCR-amplification was conducted for the five samples targeting the V3–V4 sections of the 16S-rDNA, followed by the agarose gel electrophoresis analysis to confirm the specific amplification of the sample DNA. The alpha diversity can reflect the abundance and diversity of microbial communities, as in Table 4.

The OTU value, referring to the richness, ranged between 2496 and 3531 for the five samples. The Chao, reflecting the abundance of the community, were all above 10,000. These indicated the good microbial richness in all the samples. The coverage of each sample was above 90%, indicating that the sequencing results were satisfied. The Shannon value was used to estimate the microbial diversity in the sample. The Shannon were ranked as S4 > S3 > S2 > S1 > S5, indicating that the microbial diversity of the aerobic MBR membrane tank was the best. The microbial diversity in the activated sludge could promote the removal of organics and the nitrification of ammonia nitrogen and verified the majority contribution to the degradation of organic ammonia and nitrogen in the aerobic membrane tank in Section 3.1. The Simpson refers to the uniformity and diversity of the species in the samples and little difference was found among the samples, indicating that the activated sludge was stable in the anaerobic tank, the anoxic tank and the aerobic MBR tank, guaranteeing the stable performance of nutrient removal in the system.

#### 3.2.2. Analysis of Microbial Community

The main factors and the intrinsic relationships that influence the structures and functions of the microbial community can be found via the study of the microbial community in the activated sludge in the reactor. The amplification results indicate that there were significant differences in the distribution of dominant genus at the genus level among these samples, and the distribution of dominant genus of the raw water at the genus level, as shown in Figure 5, and for the anaerobic tank, the anoxic tank, the aerobic MBR tank, and the disinfection tank, in Figure 6. It can be seen in Figure 5 that there are 20 microbial genera identified from the raw water. The dominant genera included the *Acinetobacter*, which is a kind of Gram-negative free-living saprophytic bacilli, widespread in the wastewater; the *Enterobacter*, which is a kind of Gram-negative non-pathogenic or conditionally pathogenic bacilli, widespread in the intestines of humans and animals, wastewater and soil; the *Pseudomonas*, which is a kind of organic heterotrophic bacteria, with 29 species being identified nowadays, widespread in domestic wastewater; and also *Enhydrobacter* and *Prevotella*; and others of 15.74% were unclassified genera.

As shown in Figure 6, the anaerobic tank and the anoxic tank had highly similar distribution of microbial genera, with the dominant genera of *Kofleria*, *Nannocystis,* and *Nitrospira*, with the majority being denitrifying microorganisms [18]. *Flavobacterium* and *Cloacibacterium* are denitrifying bacteria [19,20] whose high abundance in the A^2^/O-MBR process can promote denitrification. The *Ferruginibacter* was found in the aerobic tank, which is a kind of heterotrophic bacteria capable of degrading organics [21], which guarantees the degradation of the organics in the aerobic tank. In addition, *Ferruginibacter* is a kind of functional bacteria capable of synthesizing and secreting EPS [22,23], *Zoogloea* is a kind of functional bacteria capable of degrading various organics and forming the *Zoogloea* structure with the flocculating activity [19]. In this way, the organics in the water can be consumed for microbial proliferation.

### 3.3. System Membrane Fouling and Membrane Cleaning Methods

#### 3.3.1. Characterization of Membrane Fouling

In order to characterize membrane fouling, the surface of the membrane at operating time t = 0 d and t = 30 d was scanned with the electron microscopy separately (see Figure 7), in which (a) is of a new membrane with a smooth surface and evenly distributed pores while (b) is of a fouled membrane at operating time t = 30 d, covered by the fouling layer with the pores blocked by a large number of macromolecular, resulting in reduced flux.

Studies have indicated that microbial contamination is one of the main causes of membrane fouling, and there are two major categories, the extracellular polymers (EPS) and soluble microbial products (SMPs). EPS is an important contributor to activated sludge, whose large molecular size contributes to membrane fouling, resistance to degradation, and easy accumulation in the system. It leads to the increasing of sludge concentration and viscosity and the risk of membrane fouling. SMP can be divided into low molecular soluble products and high molecular soluble products according to molecular weight. The low molecular soluble products SMPs are mainly polypeptides with the general molecular weights below 1000 Da, not easy to accumulate in the system but absorbable on the internal and external surfaces of the membrane pores, resulting in blockage and fouling. The high molecular soluble products SMPs are mainly polypeptides and proteins with the molecular weights above 50 kDa, easy to accumulate in the system, and play an important role in the flocculation and sedimentation of the sludge and mainly absorbable on the external surfaces of the membrane pores and involved in the formation of the gel layer and the filter cake layer.

The transmembrane pressure difference (TMP) is a parameter of membrane fouling. The TMP changes of the membrane tank is shown in Figure 8. The device was operated for 60 d, and the trend of the transmembrane pressure difference with time indicates that the membrane fouling cycle is stable at around 30 d. At the initial stage (t = 0–10 d), the TMP rose slowly. With the gradual increase of the MLSS in the reactor, the rising rate of TMP was faster, fitting to the power function, and when the operating time was 30 d, it reached the peak value of 43 kPa. At this time, the membrane fouling was most serious and required cleaning.

#### 3.3.2. Membrane Cleaning Methods

The purpose of membrane cleaning is to remove the gel layer and filter cake layer formed on the surface of the membrane and to remove the pollutant adsorbed on the surface and inside of the membrane. Since the organic pollutant and the microbial pollutant are the main types of pollutants on the surface of the membrane, chemical cleaning should be used on the system.

The commonly used chemical agents for membrane cleaning include acidic cleaning agents, alkaline cleaning agents, oxidizing agents, complexing agents, and detergents. The acidic cleaning agents promote the dissolution of the inorganic salt ions by lowering the pH of the mixture around the membrane, and then the fouling layers deposited on the membrane pores and the membrane surface can be removed, such as carbonates, hydroxides, iron oxides, and metal sulfides. In addition, hydrolysis can occur to the proteins, polysaccharides and other high molecular organics with the acidic cleaning agents. For the effects of the alkaline cleaning agents, the removal of the organic and microbial pollutants such as proteins, oils, cellulose and humic acid mainly depends on the hydrolysis and ionization of the carboxyl groups and hydroxyl groups, so as to destroy, emulsify, disperse, and shed off the surface gel layer structure on the surface of the membrane. 

The test was carried out in two cycles; each cycle was 30 d. (The purpose of repeating the experiment is to verify the periodicity.) Membrane cleaning steps: First, remove the pipeline on the membrane module, clean the inside of the membrane module with high-pressure water, remove the activated sludge attached to the surface, and then immerse the membrane module in a cleaning tank filled with a specific cleaning solution, stirring once every 30 min, and taking out and measuring the membrane flux after 12 h.

The common oxidizing agents include sodium hypochlorite and hydrogen peroxide to create an oxidation reaction; the organics generate electronegative and hydrophilic functional groups such as ketone, aldehyde and carboxylic acid so as to loosen the structure of the fouling layer, enhance the solubility, and reduce the adsorption of various organic pollutants on the membrane, which are critical to the restoration of the membrane flux; hence the oxidizing agents and the alkaline cleaning agents are usually used in combination. Therefore, the membrane fouling was cleaned using the 6 methods in Table 5: sodium hypochlorite, sodium hypochlorite and sodium hydroxide, sodium hydroxide, hydrochloric acid, sodium hydroxide followed by hydrochloric acid, and hydrochloric acid followed by sodium hydroxide.

The restoration of the filtration performance of the membrane after different cleaning methods in Table 5 is shown in Figure 9, where the ordinate (K/K_0_) is the restoration ratio of the membrane specific flux, i.e., the ratio of the membrane flux after cleaning to that before cleaning. The cleaning effects of the combined cleaning agents were better than that of the single cleaning agent, and the restoration ratios of the membrane flux are ranked as follows: sodium hydroxide followed by hydrochloric acid > sodium hypochlorite and sodium hydroxide > hydrochloric acid followed by sodium hydroxide. The different orders of acid cleaning and alkali cleaning led to different effects on the membrane cleaning, and alkali cleaning followed by acid cleaning is the better choice. The sodium hydroxide followed by hydrochloric acid provides the best membrane flux restoration, with the ratio of 83%, which might be due to the major membrane pollutants being microbial metabolites and organics, and the inorganics being basically wrapped by EPS and SMP, the acid cleaning cannot touch the inorganic pollutants directly without alkali cleaning. The leading alkali cleaning removes the blockage on the membrane surface and pores, exposes the internal inorganics, and the following acid cleaning exerts the desired effects. The second best membrane cleaning is the sodium hypochlorite and sodium hydroxide cleaning, in which the sodium hypochlorite oxidizes the organics in the gel layer of the membrane pollutants and loosens the gel layer structure, so it is more conducive for the penetration of sodium hydroxide into the interior of the membrane pollutants to strengthen membrane cleaning effects [24,25].

The electron microscopy scanning photo of the cleaned membrane is shown in Figure 10, with sodium hydroxide followed by hydrochloric acid. The pores on the membrane are clearly visible, a large amount of pollutants on the surface of the membrane were cleaned, and only a few small and firmly attached block-like pollutants remain on the surface of the membrane. The cleaning effect of the six cleaning methods used in the test were not satisfactory and the cleaning method with the best effect still failed to make the membrane reuse. Therefore, in the following research, the cleaning method will be further developed to seek better cleaning results.

## 4. Conclusions

The wastewater discharged from public buildings was simulated by artificial water distribution and treated by the A^2^/O-MBR process. The wastewater purifying effect, the change of internal microbial community structure and MBR membrane fouling, the cleaning methods, and the cleaning performance were analyzed. 

(1) The effluent water quality of domestic wastewater treated with A^2^/O-MBR process was stable and basically met the Class IV requirements of *Environmental Quality Standards for Surface Water* in terms of COD_Cr_, BOD_5_, NH_4_^+^-N (suitable for industrial water area and entertainment water area with no direct contact with human body), and the TN and TP in the effluent met the requirements of *The Reuse of Urban Recycling Water-Water Quality Standard for Scenic Environment Use* (TN ≤ 15 mg/L, TP ≤ 1 mg/L).

(2) The microbial diversity of the raw water, the anaerobic tank, the anoxic tank, the aerobic MBR tank, and the disinfection tank were all good, and the distribution of microbial genera in the anaerobic tank and the anoxic tank were similar, with the dominant genera of *Kofleria*, *Nannocystis*, and *Nitrospira*, which are all denitrifying microorganisms. The microbial diversity of the aerobic MBR membrane tank was the best. The good microbial diversity in the activated sludge intercepted by the membrane promotes the removal of organics and the nitrification of ammonia nitrogen.

(3) The membrane was blocked with the large molecular pollutants with reduced flux, and the pollutants were mainly organics such as polysaccharides and proteins along with a small amount of inorganic salts. The combined cleaning agents were better than the single cleaning agent due to the mutual promotion effect. As the best choice, the cleaning method of sodium hydroxide followed by hydrochloric acid achieved a membrane flux restoration ratio above 80% after cleaning. However, the best cleaning method still produced 17% irreversible fouling, and these contaminated films can hardly be reused. Therefore, we need better membrane cleaning methods in order to seek better cleaning results.

## Figures and Tables

**Figure 1 membranes-09-00172-f001:**
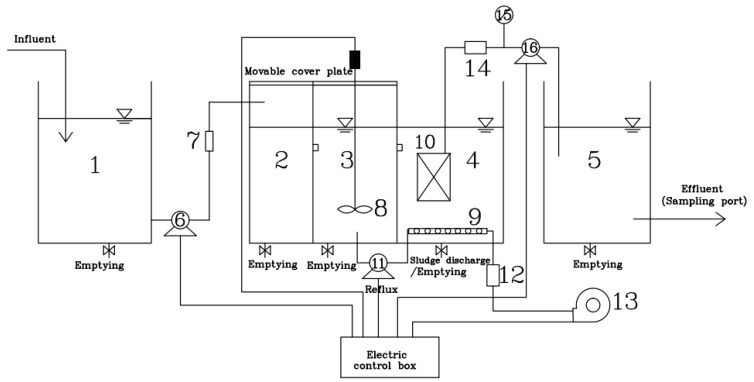
Diagram of domestic wastewater treatment devices. 1, inlet sump; 2, anaerobic tank; 3, anoxic tank; 4, aerobic MBR membrane tank; 5, disinfection tank; 6, lift pump (Shanghai Tongyuan Pump Industry Co., Ltd., Shanghai, China); 7, liquid flow meter (Hangzhou Liance Automation Technology Co., Ltd., Hangzhou, China); 8, stirrer (Shanghai Changken Test Equipment Co., Ltd., Shanghai, China); 9, aeration equipment (Shanghai Changken Test Equipment Co., Ltd., Shanghai, China); 10, plate membrane module; 11, self-priming pump (Shanghai Tongyuan Pump Industry Co., Ltd., Shanghai, China); 12, gas flowmeter (Hangzhou Liance Automation Technology Co., Ltd., Hangzhou, China); 13, air pump (Shanghai Tongyuan Pump Industry Co., Ltd., Shanghai, China); 14, liquid flowmeter (Hangzhou Liance Automation Technology Co., Ltd., Hangzhou, China); 15, vacuum pressure gauge (Shanghai Longlv Electronic Technology Co., Ltd., Shanghai, China); 16, water production pump (Shanghai Tongyuan Pump Industry Co., Ltd., Shanghai, China).

**Figure 2 membranes-09-00172-f002:**
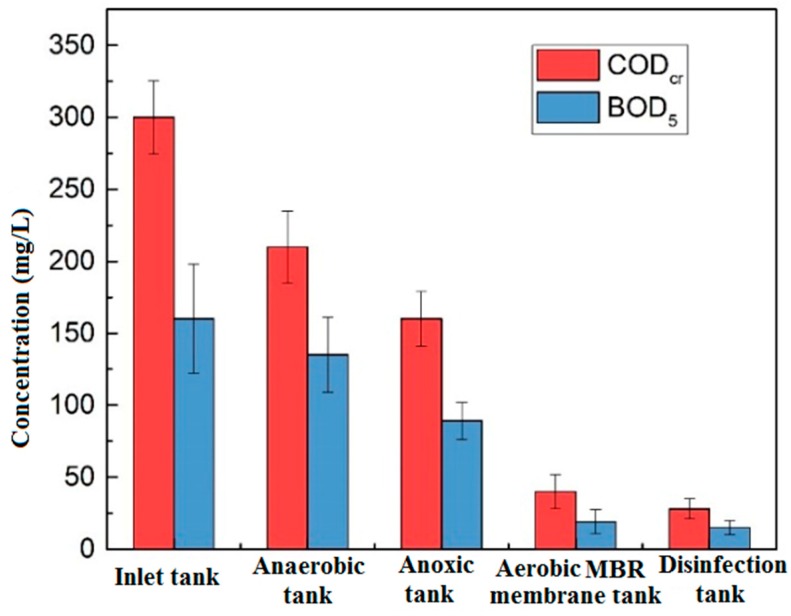
Changes of organics along the process.

**Figure 3 membranes-09-00172-f003:**
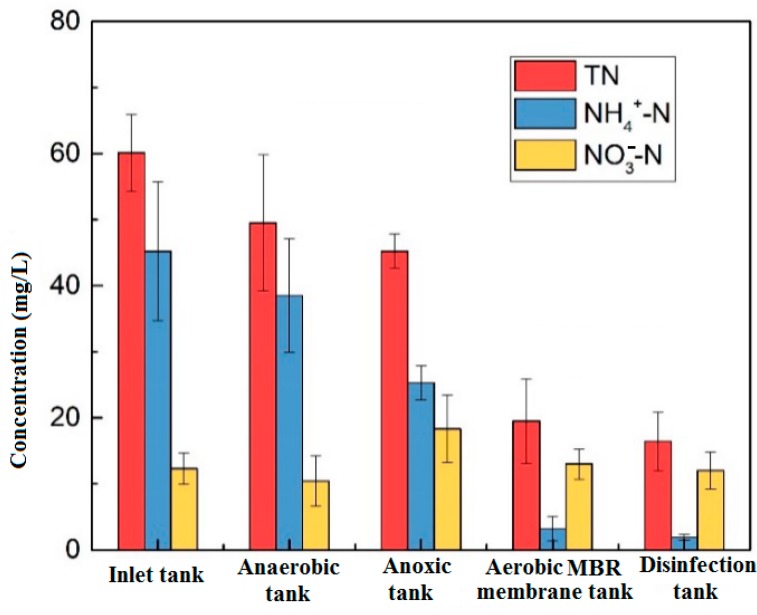
Changes of nitrogen along the process.

**Figure 4 membranes-09-00172-f004:**
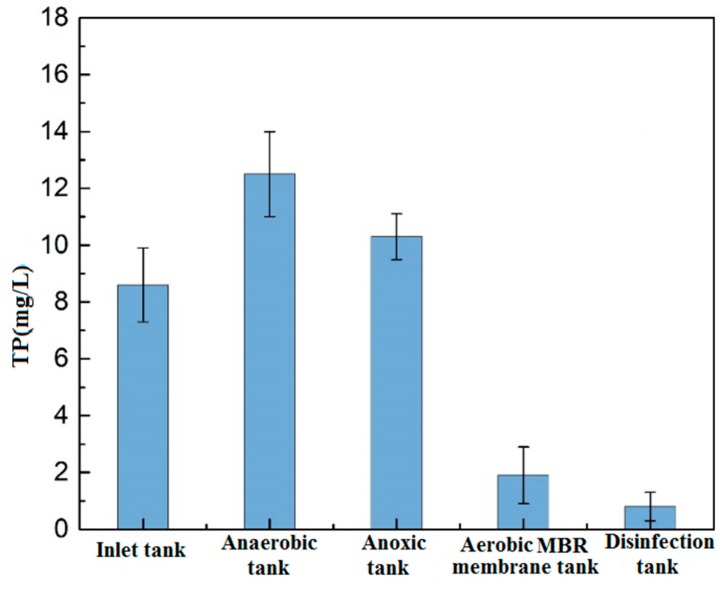
Changes of phosphorus along the process.

**Figure 5 membranes-09-00172-f005:**
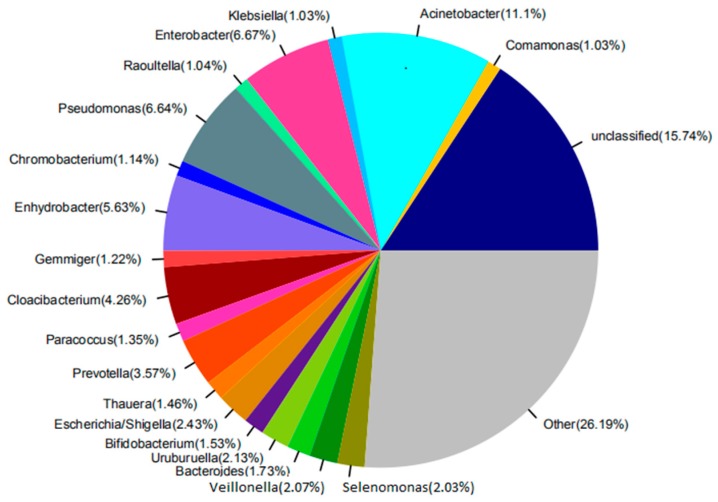
The abundance of the raw water at the genus level.

**Figure 6 membranes-09-00172-f006:**
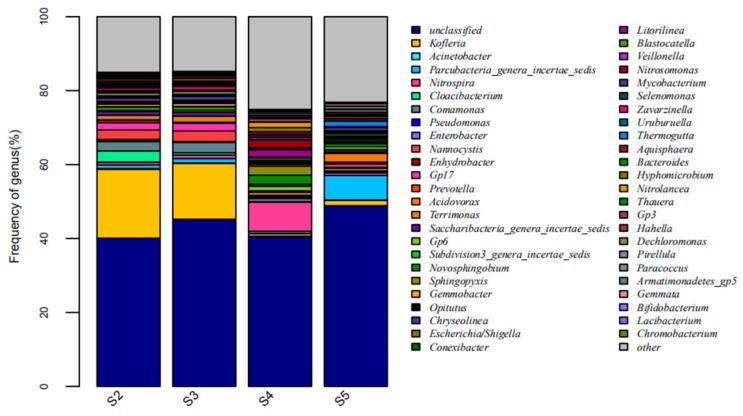
The abundance of the anaerobic tank, the anoxic tank, the aerobic MBR tank, and the disinfection tank at the genus level.

**Figure 7 membranes-09-00172-f007:**
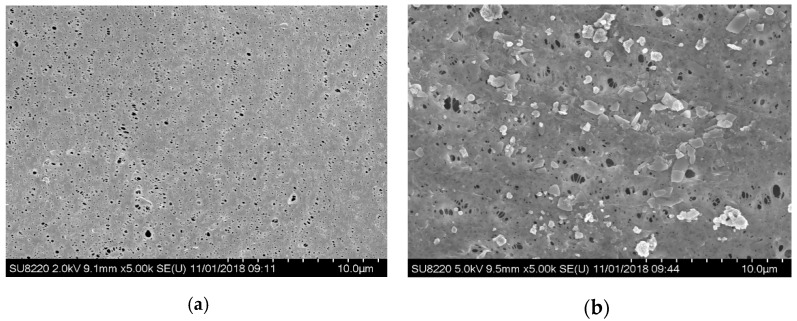
SEM characterization of MBR membrane fouling. (**a**) SEM of the surface of the new membrane; (**b**) SEM of the fouled membrane surface after use.

**Figure 8 membranes-09-00172-f008:**
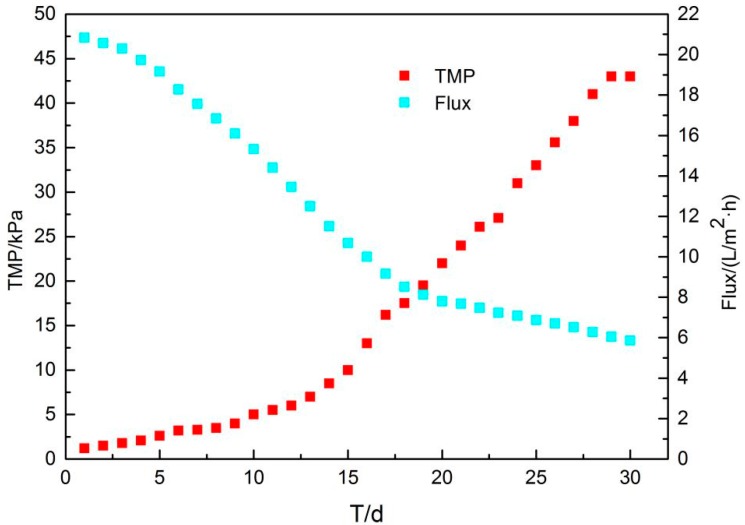
TMP and flux changes with time.

**Figure 9 membranes-09-00172-f009:**
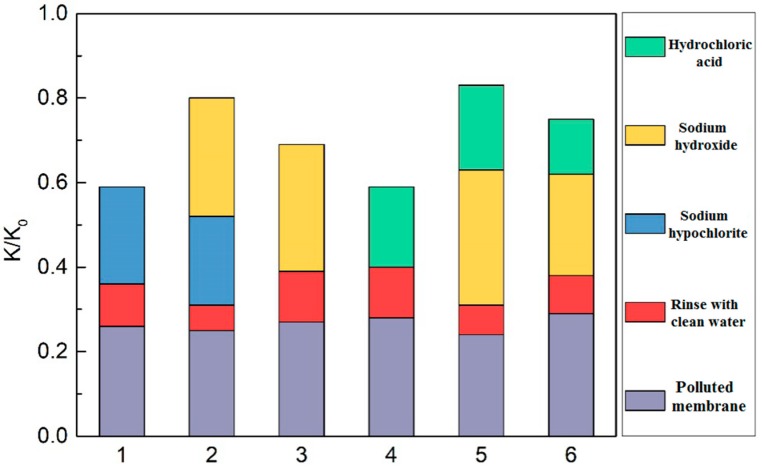
Effects of different cleaning methods for membrane fouling.

**Figure 10 membranes-09-00172-f010:**
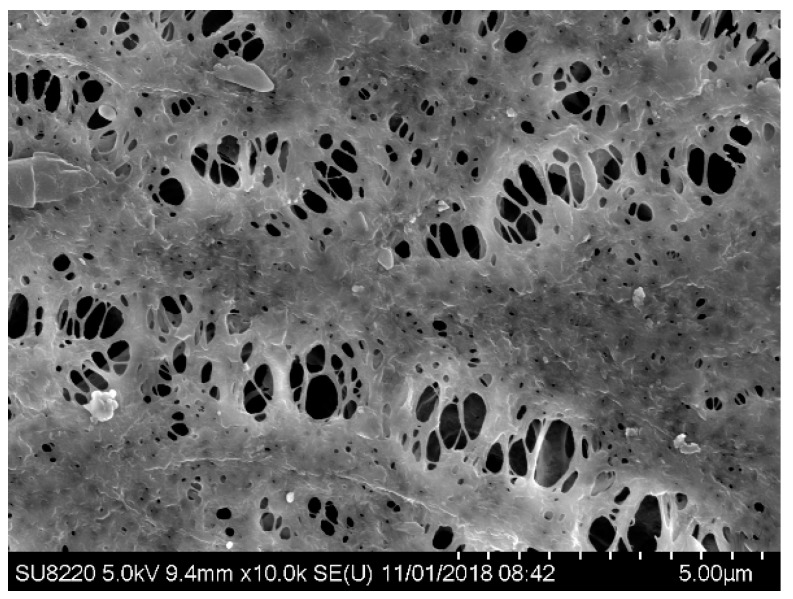
SEM image of the MBR membrane after cleaning.

**Table 1 membranes-09-00172-t001:** Raw water quality.

Parameter	Concentration (mg/L)
COD_Cr_	300 ± 25.3
BOD_5_	162 ± 38
NH_4_^+^-N	45.2 ± 13.5
TN	60.1 ± 5.8
TP	8.6 ± 4.8

**Table 2 membranes-09-00172-t002:** The operation parameters of the device.

Operation Parameter	Value
Hydraulic retention time (HRT)	Anaerobic tank/1.5 h; Anoxic tank/2 h; Aerobic tank/4 h
Dissolved oxygen (DO)	Anaerobic tank < 0.2 mg/L; Anoxic tank 0.2–0.4 mg/L; Aerobic tank 3 mg/L
Sludge retention time (SRT)	30 d
Mixed liquid suspended solids (MLSS)	7000–9000 mg/L
Flow rate	6 L/min
The dimension of reactors	Anaerobic tank/80 L; Anoxic tank/80 L; Aerobic tank/200 L
Temperature	26 ± 6 °C

**Table 3 membranes-09-00172-t003:** MBR plate membrane performance parameters.

Parameter	Value
Membrane material	Polyvinylidene fluoride (PVDF)
Membrane form	Immersed plate microfiltration membrane
Pore size	0.20 μm
Diaphragm size	1570 mm × 500 mm × 7 mm
Effective membrane area	1.4 m^2^
Design running flux	10–25 L/m^2^·h
Applicable cleaning method	Pickling cleaning, alkaline cleaning, and physical cleaning

**Table 4 membranes-09-00172-t004:** Abundance and diversity indexes of microbial communities in the samples.

Sample	Sample Name	OTU	Chao	Coverage	Shannon	Simpson
S1	Influent	3388	18,174.27	0.93	5.12	0.03
S2	Anaerobic tank	3431	19,438.27	0.92	5.19	0.04
S3	Anoxic tank	3531	16,194.30	0.94	5.22	0.03
S4	Aerobic MBR tank	3128	18,060.14	0.94	5.95	0.02
S5	Disinfection tank	2496	15,483.53	0.92	4.31	0.01

**Table 5 membranes-09-00172-t005:** Different membrane cleaning methods.

Number	Cleaning Agents	Mass Concentration
1	Sodium hypochlorite	0.3%
2	Sodium hypochlorite & sodium hydroxide	0.3 + 0.1%
3	Sodium hydroxide	0.1%
4	Hydrochloric acid	0.2%
5	Sodium hydroxide followed by hydrochloric acid	0.1 + 0.2%
6	Hydrochloric acid followed by sodium hydroxide	0.2 + 0.1%

Note: 1. The cleaning temperature is 25 °C. 2. The above cleaning steps are first washed with clean water, and then cleaned with chemicals.

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
