# Peer review of "Research on Operation Efficiency and Membrane Fouling of A2/O-MBR in Reclaimed Water Treatment"

_membranes, 2019, doi:10.3390/membranes9120172_

Round 1

Reviewer 1 Report

The article “Research on Operation efficiency and Membrane Fouing of A2/O-MBR in Reclaimed Water Treatment” indicated the removal efficiency of pollutants by an A2O-MBR system, and the microbial community in reactors, the methods for membrane cleaning. In general, the A2O-MBR system has been investigate in many studies. Compared to some studies, no new findings are found in this manuscript. Besides, there are some serious problems in this study as belows:

The Introduction section is weak and insufficient in terms of theory. Lines 140-157, 224-237, 251-266 should be located in the Introduction section with citations. The Materials and Methods section is not clear and lack of information on the experiment. The methodology did not show several important operational parameters such as hydraulic retention time (HRT), sludge retention time (SRT), dissolved oxygen (DO) concentration, mixed liquor suspended solids (MLSS), flow rate, operational flux, the dimension of reactors, sampling locations. The chemicals used to make artificial wastewater should be presented. As shown in Figure 1, the input and output of the anaerobic tank were near the water level, resulting in reducing the contact between wastewater and anaerobic bacteria. Similarly, there is no comment on how to mix wastewater and disinfectant in each tank. The influent concentration of COD of 300 mg/L was treated with the system including anaerobic, anoxic, MBR tanks. However, the effluent concentration was at 30 mg/L, which was relatively higher than that of usual wastewater treatment processes. This indicated that the system was configured improperly or operated ineffectively. In addition, the COD concentrations were around 300 and 150 mg/L in the influent and the anoxic tank, respectively (Figure 2). These data indicated that half of COD was removed by anaerobic and anoxic tanks. The authors concluded that the aerobic tank accounted for 75% in the COD removal, it is not correct. In section 3.2, why did microorganisms present in the artificial wastewater (the influent of this study)? The OTU value of the disinfection tank was quite high (Table 3) which showed that the disinfection efficiency was not good. The disinfectant concentration and disinfection time were not presented in this study. Besides, some bacteria such as Thiothrix, Flavobacterium, Ferruginibacter, Zoogloea (lines 199-212) were not found in Figure 6. Microorganisms involved in phosphorus removal were not also mentioned. How to clean a membrane module by six methods as shown in Table 4 while one membrane module was used in this study? Was the membrane module cleaned inside or outside the system? How long? The authors should compare the results with other studies to highlight the new findings of this study.

Reviewer 2 Report

The manuscript is written well and clearly. There are some concerns which the authors need to respond to and correct in the manuscript. 

General comments that need to be addressed in the manuscript by justifying:

The authors need to discuss the selection of their wastewater characteristics in their manuscript. What is the C: N:P ratio of the simulated wastewater? Does it represent the normal wastewater in Beijing? If yes, cite it. If no, justify in the text why this simulated wastewater characteristic was taken. What is the necessitates the introduction of the A²/O-MBR process with the wastewater characteristics mentioned in Table 1. If the C:N:P ratio is in the range 100:10:1 to 100:5:1 the biomass should be able to degrade the nutrients without much trouble and as normal MBR’s the effluent was not able to meet the Phosphorous standards.  here did the authors collect the sludge to start the overall experimental setup?  Based on the results from COD, TN and TP the anaerobic tank doesn’t seem to be necessary as the normal MBR process with the Anoxic tank is doing most of the work. Also, the microbial colonies seem to similar between them. Can the authors justify the need for an anaerobic tank as it seems like another anoxic tank and also how do the authors maintain the anaerobic conditions? is the tank sealed, was methane being produced?  

Some minor corrections and comments that need an answer from the authors:

Correct English: Line 63, 64 Add BOD5 data in table 1. Results section shows BOD5 was measured.   hinese Water and Wastewater Monitoring Methods, 4th ed similar to the APHA standard methods for water and wastewater treatment? What is different? What are Class IV requirements of “Environmental Quality Standards for Surface Water”? Can the authors give some numbers on the discharge guidelines for the parameters that they have evaluated ?
What is the oxygen level in the anoxic tank and the MBR tank? Mention in the method section.
Can the authors describe in more detail the experimental setup in the manuscript? For example, what is the difference b/w the anaerobic, anoxic tank. Is the anaerobic tank hermetically sealed? What does the disinfection tank do? What is inside? how does it disinfect? It is better to mention in the methods, operational strategies, how long was the experimental run and when the sampling was done, etc Fig 2: What is the inlet tank doing for the COD and BOD. It seems just keeping water in the inlet tank reduces the COD by 100 mg/L and the difference b/w using anaerobic and anoxic tank is not much different w.r.t COD. L115: “The removal of CODcr mainly contributed by the aerobic tank (accounting for 75%)”. The reviewer is not so sure about this statement. As Fig 2 based on rough calculations it seems, inlet tank removes 25 % and around 30 % b/w the anaerobic and anoxic tank and anoxic and MBR tank around 75 %.  L152: what do the authors mean by “concersentration” and “excist”? Section 3.2: When was the analysis conducted? Do the authors have the same data of the sludge that was used to start the A²/0-MBR process? L199: The sentence makes no sense. “As shown in figure 6, the anaerobic tank and the anaerobic tank had highly similar distribution” L204: As the authors have yet to speak about fouling, this sentence seems completely out of place. “Therefore, the high abundance of Thiothrix in the microbial community in the A2/O-MBR system might be one of the reason of the serious membrane fouling.” L207: Better to use “strongly” instead of “stengthenly”.  Fig 8: Can the authors show the flux data too on the same graph on the secondary axis? As it raises the question of how do the authors decide 43 kPA is the time for cleaning as the flux is not presented its difficult to evaluate?  L266: Typographical error: Therefor, also the sentence L266 -267 is not clear. Section 3.3.2: is an interesting section But how did the authors clean the membrane 6 times to produce these results as the membrane fouled 2 times as per Fig 8 TMP data in 60 days? Which membranes are being used and how are they fouled? The authors need to make this section clearer. What is the general cleaning protocol used in this manuscript? The authors are requested to elaborate on the steps and time taken to conduct each cleaning procedure.
The authors need to compare the microbial community with other literature in the results before concluding that it is good. 80 % cleaning efficiency seems not to be great as it indicates 20 % of the pores on the membrane are irreversible fouled? Normally, irreversible fouling should be less than 1 % and it is the reviewer’s option that cleaning protocols are not conducted thoroughly, therefore, the cleaning efficiency is lower. The authors are advised to do more systematic cleaning protocol where they can use the membrane permeability to classify the various fouling mechanisms ( reversible fouling, recoverable fouling, irreversible fouling, etc)

Overall:

The authors need to work a bit more on presenting their results in the manuscript and describe their methodology clearly for the readers of their work.

Reviewer 3 Report

The manuscript has presented the development of an integrated process based on A2/O MBR processes for the treatment of domestic wastewater. The paper analyzed the wastewater purifying effect, the change of internal microbial community structure, MBR membrane fouling, the cleaning methods and the cleaning performance.  I would recommend publication of the manuscript in “Membranes” after some minor edits. The remarks are provided as follow:

The authors should specify the meaning of the abbreviations when appear for the first time in the text (MLSS, A2/O, TN and TP, etc.) The authors should improve the resolution of the legend of Figure 6 Line 29, change “showd” to “showed” Line 129, change “emmission” to “emission” Line 152, change “concersentration” to “concentration” Line 154, regarding “excist”, do you mean “exist”? please check it Lines 189-190, 199, please change “figure” to “Figure” Line 207, regarding “strengthenly”, do you mean “strengthened”? please check it Line 266, regarding“therefor” do you mean “therefore” please chek it Line 292, change “scaning” to “scanning” Paragraph 2, the authors should briefly describe the methods used I suggest changing in the table 2 “average aperture” to “pore size” Paragraph 3.3.1, the authors should also discuss the performance of MBR in terms of the permeate flux. Paragraph 3.3.2, the authors should add the operation time for washing step of membrane

Round 2

Reviewer 1 Report

The manuscript has been improved in terms of contents.

Please, chek the conclusion part.

The conclusive mark on your research will be required according to the objectives.

Author Response

Point 1: Please, check the conclusion part. The conclusive mark on your research will be required according to the objectives.

Response 1: Thank you for your advice, this part has been added to the conclusion of the text.

Round 2

The authors have made effort to improve their manuscript but this version of the manuscript had its shortcomings. Below are some questions that the authors need to address.

Comments:

L60 and L70: what does membrane pollution mean? Fouling? if yes, then use the good term for the phenomenon. If no, then explain better.  L82-84: Authors are avoiding the question of why this wastewater proportion was selected and what is the CNP ratio that necessitates the A2/0-MBR system. The authors did not answer the question previously and coupled my question to other questions and answered only one of the questions posed. (Reference 2nd question from Round 1 of the reviewer) Table 2: Mention the water temperature range too. L128: Correction: "See eq. 2.1" and also label the equation.
In Figures 2, 3 and 4 add the stringent requirements of effluent water so that the readers can see and compare. The authors forgot to mention in the manuscript but did mention in the reviewer's reply.  Figure 8: Authors have forgotten to add the flux data on the secondary axis again. L301: The authors are saying 1 cycle last 30 days and they had 6 cycles then the membrane operated for 180 days and data is only presented for 60 days. It is really confusing how the authors did the cleaning 6 times. The reviewers want to know, how many times did the authors do the membrane cleaning? 2 times? or 6 times. And if its 6 times how and where is the flux data to show that 6 times cleaning was necessary for 60 days.  L318: at the end of the paragraph the need to add the information that all the cleaning methods that they have tested are not good enough at the present moment as their best cleaning techniques still lets 17 % of irreversible fouling i.e. after 5 cycles, the membranes will be almost unusable even with their best cleaning technique. Irreversible fouling should be less than 1 - 3 % and the authors can find literature that supports this data. Avoid using “remarkable cleaning effects” etc. As the overall cleaning efficiency is not good. It is OK to mention the cleaning protocols need further r&d and the current approach is not suitable with A2/0-MBR. 

Round 3

Reviewer 2 Report

The authors have addressed my comments and modified the manuscript. Its in its acceptable format.

Just one comment L365: Authors are requested to use their own words to describe that the cleaning methods are not viable at this point. I wrote to the authors to illustrate the fact that there is a problem in cleaning but wrote in a running language therefore it would be better not to use my exact words in your manuscript.